# Alterations in Whey Protein Abundance Correlated with the Somatic Cell Count Identified via Label-Free and Selected Reaction Monitoring Proteomic Approaches

**DOI:** 10.3390/ani15050675

**Published:** 2025-02-26

**Authors:** Jing Li, Kaixu Chen, Changjiang Zang, Xiaowei Zhao, Zhiqiang Cheng, Xiaobin Li, Caidie Wang, Yong Chen, Kailun Yang

**Affiliations:** 1Xinjiang Key Laboratory of Herbivore Nutrition for Meat & Milk, College of Animal Science, Xinjiang Agricultural University, Urumqi 830052, China; jing03312024@163.com (J.L.); chenkaixu@xjau.edu.cn (K.C.); xiaowei1986mm@163.com (X.Z.); cheng07162022@163.com (Z.C.); lxb262819@163.com (X.L.); wcd@xjau.edu.cn (C.W.); xjaucy@163.com (Y.C.); yangkailun2002@aliyun.com (K.Y.); 2Institute of Animal Science and Veterinary Medicine, Anhui Academy of Agricultural Sciences, Hefei 230031, China

**Keywords:** somatic cell count, milk, whey protein, proteomics

## Abstract

The somatic cell count (SCC) in milk is widely recognized as a reliable indicator for assessing udder health and milk quality. However, the precise relationship between the SCC and milk quality remains inadequately understood. In this study, we conducted a comprehensive analysis of milk samples with varying SCCs to better understand the correlation between specific whey protein levels, the SCC, and intramammary infections. Transition changes in whey protein components dependent on the milk SCC were observed, and several proteins were found to be correlated with SCC. These findings provide a better rudimentary understanding of how milk quality corresponds to the SCC.

## 1. Introduction

The somatic cell count (SCC) in milk is considered to be a gold standard parameter for monitoring udder health and milk quality in the dairy industry [1]. It represents the number of somatic cells, primarily leukocytes, present in a milliliter of milk. The SCC typically increases in response to infection or inflammation of the udder, often triggered by pathogenic microorganisms. Elevated SCC levels are commonly associated with a decline in milk quality, as they can lead to alterations in milk composition, such as reduced casein content and an increase in whey proteins. Monitoring SCC is essential for maintaining udder health and ensuring milk quality. To date, numerous studies have investigated the correlation between milk components and SCC, aiming to establish a threshold for assessing intramammary infections and milk quality. Most of these previous studies have consistently indicated that an SCC threshold of 20 × 10^4^ cells/mL is commonly recognized as indicative of mastitis occurrence [2,3]. Other studies have demonstrated that an SCC threshold of 10 × 10^4^ cells/mL is sufficiently sensitive for detecting pathogen infection in cows [4,5]. Furthermore, SCC serves as a critical tool for detecting and managing mastitis, a condition that can significantly impact milk yield and farm profitability. As a result, many countries have established milk SCC threshold as a regulatory measure to monitor and maintain milk quality. 

Specifically, the SCC threshold varies across different countries, with the European Union and Canada setting it at 40 × 10^4^ cells/mL [6,7], New Zealand and Australia at 20 × 10^4^ cells/mL, and the United States at 75 × 10^4^ cells/mL [8]. However, the relationship between the milk SCC threshold and milk composition has not been clearly characterized. In general, an increase in the SCC is associated with a decrease in milk yield, caseins, and lactose content, as well as an increase in whey protein content [9,10]. Moreover, the coagulation properties and processing properties of milk are also affected by the SCC [11]. Thus, it is imperative to pay close attention to the correlation between milk composition and the SCC. 

Numerous studies have probed the correlation between protein content and the SCC, given that milk proteins are a primary source of bioactive substances [12,13]. These studies discovered that rising SCC in milk was linked with increased levels of immune-related proteins and enzymes in milk whey, while there was a concurrent degradation of caseins [13,14]. For this purpose, a range of technologies, including ELISA, gel electrophoresis, and proteomics approaches, have been employed [15]. Specifically, with the advancement of omics technologies, proteomic approaches allow for the investigation of the relationship between alterations in the milk proteins and somatic cell count at the global protein level. To examine the alterations in protein constituents at the proteome level, comparative analyses of bovine milk proteins from healthy cows and those with experimentally induced mastitis have been conducted in various studies [15]. Additionally, untargeted label-free and labeled quantitative proteomic analyses have investigated milk proteins from both healthy and naturally infected mastitic cows, using SCC as a metric [16,17]. As well, targeted proteomics based on selected reaction monitoring/multiple reaction monitoring methods has emerged as a reliable approach for validating MS-based proteomics data. In a study in which 40 peptides were simultaneously monitored, several peptides from caseins and β-lactoglobulin in milk samples from Holstein cows were significantly affected by subclinical mastitis induced by *Streptococcus agalactiae* and *Prototheca* spp. [18]. Interestingly, the application of microbiome and metabolomics analysis has unveiled variations in the microbiology and metabolites present in the milk of cows with varying SCCs, suggesting that certain metabolites and microorganisms are related to milk SCC [19]. This study investigates the changes in milk constituents associated with SCC, representing a preliminary step toward a deeper comprehension of the correlation between milk proteins and SCC. It offers valuable insights into the potential effects of SCC on milk quality, providing a foundation for further research into the molecular mechanisms underlying SCC-related alterations in milk composition.

## 2. Materials and Methods

### 2.1. Sample Collection

The selected farm houses over 600 lactating Chinese Holstein cows, located near Chuzhou City in Anhui Province, China, with DHI testing conducted on milk samples 1–2 times per month. Prior to sample collection, a preliminary selection of cows was performed based on DHI data, focusing on cows with 2–3 parities, similar milk yields, and in mid-to-late lactation. Additionally, these cows were fed a uniform diet. Subsequently, cows were milked automatically using a milking system equipped with a diverter, with milk yield recorded and milk samples collected from the diverter for further analysis. A total of 100 milk samples (50 mL × 2) were collected from individual dairy cows. The milk yield of each cow was also recorded. One portion of the collected samples was utilized to determine the protein and fat content as well as the SCC, using Fossomatic 5000 and FT120 analyzers (Foss Electric, Hillerød, Denmark). The remaining milk samples (50 mL) were frozen at −20 °C. From the initial collection, a subset of sixty milk samples was selected and divided into five different groups based on the SCC levels, with each group consisting of 12 samples. The classifications of the milk groups were as follows: 6–9 (7.2, S1 group), 17–20 (18.4, S2 group), 38–42 (40.4, S3 group), 68–80 (73.6, S4 group), and 176–243 (208.9, S5 group) × 10^4^ cells/mL. Appendix A provides information on milk yield, the SCC, and milk components for each of the five milk groups.

### 2.2. Milk Whey Separation

The milk samples were thawed at 20 °C. Upon thawing, four samples from each group were pooled in equal volumes to obtain three biological replicates. The mixed samples were then centrifuged at 3000× *g* at 4 °C for 20 min. The upper layer containing milk fat was removed, while the middle layer consisting of skimmed milk was carefully transferred into new tubes. The collection of milk whey was obtained as described in a previous study [20]. In brief, 1 mL of skimmed milk was mixed with 30 µL of 33% acetic acid and 30 µL of 3.3 M sodium acetate, followed by centrifugation at 10,000× *g* at 4 °C for 15 min. The liquid phase of the milk whey was subsequently collected, and the protein concentration was determined using a Bradford kit (Beyotime Biotechnology, Shanghai, China). In this procedure, the sample was mixed with Coomassie Brilliant Blue G-250 dye, and absorbance was measured at 595 nm. The absorbance value is directly proportional to the protein concentration, with bovine serum albumin used as the standard.

### 2.3. Protein Digestion 

One hundred micrograms of whey protein was combined with a buffer solution containing 4% sodium dodecyl sulfate, 100 mM Tris-HCL, and 0.5 M dithiothreitol. The mixture was heated in water at 95 °C for 5 min. Subsequently, the protein samples were mixed with 200 μL of UT buffer (8 M urea and 150 mM Tris-HCl, pH 8.0). The resulting mixture was transferred to a filter tube with 10 kDa cutoff (Sartorius, Goettingen, Germany) and then centrifugated at 14,000× *g* for 25 min. After the samples were washed twice with UT buffer, 60 μL of a 50 mM iodoacetamide solution was added to the samples, which were then incubated at 25 °C for 45 min in the dark. Finally, the samples underwent three washes with 200 μL of UT buffer and were mixed with 60 μL of trypsin buffer (2 μg trypsin in 50 mM NH_4_HCO_3_). The mixture was then incubated at 37 °C for 16–18 h. Following this, the filter was transferred to a new tube and subjected to centrifugation at 14,000× *g* for 15 min, resulting in the collection of eluates. The filter was washed twice with 50 mM NH_4_HCO_3_, and the eluates were pooled and dried in a speed vacuum.

### 2.4. Liquid Chromatography–Tandem Mass Spectrometry Analysis

Tryptic peptides were dissolved with 0.1% formic acid and subjected to chromatography using an EASY-nLC 1200 system coupled with a Q-Exactive Plus instrument (Thermo Fisher Scientific, Waltham, MA, USA). Three pooled samples were analyzed for each group. Prior to mass spectrometry (MS) analysis, the column was equilibrated with 95% (*v*/*v*) buffer A (0.1% formic acid). The peptide mixtures were loaded onto a trap column (20 mm × 100 μm, 5 μm) using an autosampler and separated on a reversed-phase column (120 mm × 75 μm, 3 μm, Thermo Fisher Scientific) with buffer B (0.1% formic acid and 85% acetonitrile). The separation protocol consisted of the following steps: buffer B was initially adjusted from 5% to 8% for 2 min, then increased from 8% to 23% for 88 min, subsequently elevated from 23% to 40% for 10 min, followed by a gradual increase from 40% to 100% for 8 min, and finally maintained at 100% for 12 min.

The Q-Exactive instrument was operated in the positive ion mode with data-dependent acquisition analysis for 120 min. The mass range for the scanning precursor ion was set at 300–1800 mass/charge (m/z). The resolving power was established at 70,000 at m/z 200 for the MS scan and 17,500 at m/z 200 for the MS/MS scans. The automatic gain control target values were configured as 1e6 for the MS and 1e5 for the MS/MS scans. On the basis of the MS scan, the top 20 most abundant precursor ions were selected and subjected to MS/MS analysis via higher energy collisional dissociation. The isolation window was determined as 1.6 Th, the maximum ion injection time was 50 ms, and the normalized collision energy value was 27 eV. All data were acquired using Xcalibur 3.0 software (Thermo Fisher Scientific).

### 2.5. Protein Identification and Quantitation

The raw files were analyzed using Maxquant software (version 1.6.0.16). Subsequently, a search was conducted against a Bos taurus dataset downloaded from the UniProt database (46,754 entries, 12/2020). The maximum allowable number of missed cleavage sites was restricted to 2. The initial peptide search tolerance and MS/MS tolerance were both set at 20 ppm. The fixed modification was defined as carbamidomethylation of cysteine, while the variable modification included methionine oxidation and protein N-terminus acetylation. The decoy database pattern was set as the reversed version of the target database, wherein protein and peptide identification were filtered based on a false discovery rate of less than 1%. The identified proteins were filtered with at least two peptides and quantified by label-free quantification methods, specifically employing razor and unique peptides. 

### 2.6. Selected Reaction Monitoring (SRM) Validation

Eleven differential proteins (peptidoglycan recognition protein 1, lymphocyte cytosolic protein 1, leukocyte elastase inhibitor, serine dehydratase/threonine deaminase, interleukin 18 binding protein, desmin, azurocidin 1, and cathelicidin-1, -4, -5, and -6) were chosen and validated using a selected reaction monitoring approach. Peptides selected for SRM were based on label-free proteomic identification, with a focus on those that were unique to the target protein, exhibited no missed cleavage sites during digestion, were 7–20 amino acids in length, and were free from post-translational modifications. The selected peptides from each target protein were imported into Xcalibur software. Tryptic peptides (2 μg) were then subjected to chromatography on an EASY-nLC 1200 coupled with a Q-Exactive Plus instrument. The elution of the peptides was conducted using a gradient of buffer C (0.1% formic acid and 95% acetonitrile) at specific time intervals: from 3% to 5% for 2 min, from 5% to 25% for 35 min, from 25% to 45% for 8 min, from 45 to 90% for 3 min, and maintained at 90% for 12 min, at a flow of 300 nL/min. The MS was operated in the positive ion mode, with a mass range of 300–1500 mass/charge (m/z). The selection of target peptides was based on the precursor m/z, identified through the MS scan, and subjected to MS/MS analysis via higher energy collisional dissociation with a resolving power of 35,000 and a normalized collision energy of 27 eV. The isolation window was established at 1.6 Th, the maximum ion injection time was set to 50 ms, and the automatic gain control target value was configured at 3e6. The raw files were analyzed utilizing Skyline software (Version 4.1, MacCoss Lab, Seattle, WA, USA; skyline.ms/project/home/begin.view, 05/2021) to filter and integrate precursor signals of target peptides, thereby generating the transition list and calculating the peak area values.

### 2.7. Data Analysis 

The proteins derived from various milk groups were imported, categorized, and preprocessed, including log2 transformation and handling of missing values, followed by statistical analysis via ANOVA, as integrated within Perseus software (Version 2.3.1.0, www.perseus-framework.org, 06/2023), with significance determined by *p* values less than 0.05. Additionally, hierarchical clustering and principal component analysis of the quantified proteins were conducted using Perseus software. The functional analysis of the differentially abundant proteins was carried out using the DAVID Functional Annotation Tools (david.ncifcrf.gov, 02/2024). Differentially abundant proteins were analyzed for protein–protein interactions (PPI) using STRING software (string-db.org/, 01/2025).

## 3. Results

### 3.1. Proteomics Analysis of Milk Whey Among the Five Different Milk Groups

Changes in milk components and milk yield were compared among the five different milk groups. The milk protein content of the S5 group was significantly higher than the S1 group, whereas the milk yield and fat content were not different. A proteomics analysis of milk whey was then performed, and a total of 359 proteins were quantified using a label-free approach among the five different milk groups. The Pearson correlation coefficients were found to be greater than 0.92 among the three replicates for each group, showing that the data were highly reproducible (Appendix A). 

Statistical analysis of the identified whey proteins indicated that 136 proteins were differentially abundant among the studied groups (Appendix A). Of these, several proteins, such as azurocidin 1, cathelicidin–1, peptidoglycan recognition protein 1, interleukin 18 binding protein, leukocyte elastase inhibitor, transketolase, and serum amyloid A protein significantly increased from the S1 to S5 groups, whereas topoisomerase I, laminin subunit beta 1, 60S ribosomal protein L12, and desmin significantly decreased from the S1 to S5 groups. Several proteins, such as serine dehydratase/threonine deaminase, lymphocyte cytosolic protein 1, and Rho GDP-dissociation inhibitor 2 were barely detectable in the S1 and S2 groups. In addition, using a fold change of 2 or higher, and compared to the S1 group, 43 whey proteins were altered in the S2 group, and 53, 57, and 66 whey proteins were altered in the S3, S4, and S5 groups, respectively. These data clearly show that the number of differentially abundant proteins significantly increased from the S2 to S5 groups.

To confirm these changes in whey proteins among the different milk groups, 11 selected proteins were analyzed on the basis of their unique peptides using an SRM approach; information on their target peptides is listed in Table 1 and Appendix A. The relative intensities of azurocidin 1, peptidoglycan recognition protein 1, leukocyte elastase inhibitor, interleukin 18 binding protein, and cathelicidin-1, -4, -5 and -6 all significantly increased from the S1 to S5 groups, whereas desmin significantly decreased from the S1 to S3 groups and then remained constant in the S3–S5 groups. The results for these proteins obtained using the SRM approach were almost identical to the results obtained using label-free proteomics.

### 3.2. Functional Analysis of Differentially Abundant Proteins

For all the differentially abundant proteins, their biological function and cellular components were predicted according to their annotations using the DAVID functional annotation tool (Figure 1). Based on the biological process, we found that most of the differentially abundant proteins were associated with the response to stimulus, localization, and defense response. Based on the cellular component, most of the differentially abundant proteins were classified into extracellular region (including extracellular space and exosome) and vesicle (including extracellular and cytoplasmic vesicle). Moreover, differentially abundant proteins were significantly related to the complement and coagulation cascades, antigen processing and presentation, salivary secretion, protein processing in the endoplasmic reticulum, the spliceosome, and the biosynthesis of amino acid pathways (Table 2). The PPI network of 129 proteins was extracted from the String database (Figure 2). The interaction sources included text mining, experimental data, and curated databases. A minimum required interaction score of 0.7, indicating high confidence, was applied to select significant protein interactions. Four proteins, including serum albumin, haptoglobin, alpha-1-antiproteinase, and lactoferrin, were identified as hub proteins by screening the nodes with a degree of connectivity greater than 10. Notably, both haptoglobin and lactoferrin are implicated in the defense response. Details of the PPI network are provided in Appendix A. 

### 3.3. Clustering and Principal Component Analysis of Differentially Abundant Proteins

All differentially abundant milk whey proteins were analyzed by hierarchical clustering using Perseus software, as shown in Figure 3. We found that the S1 and S2 groups had a similar protein pattern and formed a subcluster, compared to S3, S4, and S5 groups, in which the S3 and S4 groups were similar to each other and formed a subcluster, and then the S5 group joined into this subcluster. A principal component analysis (PCA) of the differentially abundant whey proteins from each milk group was then performed. The score plots and loading plots of PCA are presented in Figure 3b and Appendix A. The analysis indicated that the protein patterns from each group tended to cluster together, suggesting developmental changes across the S1 to S5 groups. According to the loading plots, several proteins, including protein disulfide isomerase–associated 6, vimentin, desmin, tropomyosin-1, azurocidin 1, peptidoglycan recognition protein 1, haptoglobin, serpin B4, cathelicidin-1, and cathelicidin-2, were identified as significant contributors to the observed developmental changes from the S1 to S5 groups.

## 4. Discussion

In this study, whey protein profiles associated with varying SCCs in bovine milk were characterized using a label-free proteomics approach. Subsequent validation of differentially abundant proteins was conducted using the SRM method. Notably, several proteins, including azurocidin 1, cathelicidin-4, and peptidoglycan recognition protein 1, exhibited significant increases from the S1 to S5 groups. Conversely, proteins like serine dehydratase/threonine deaminase and lymphocyte cytosolic protein 1 were scarcely detectable in the S1 and S2 groups. Consequently, these proteins could potentially serve as indicators of intramammary infections and be utilized for assessing milk quality. Based on the protein annotations, alterations in the whey proteins were primarily linked to functions such as antibacterial activity, enzymes, inflammatory response, acute phase response, complementary components, and the cytoskeleton.

A majority of proteins that showed an increase with SCC were implicated in antibacterial activities. In addition to the already known cathelicidins, peptidoglycan recognition protein 1, leukocyte elastase inhibitor, and interleukin 18 binding protein, novel disease-related proteins in bovine milk, were identified as azurocidin 1, β-defensin 2, and β-defensin 12. Previous investigations have also reported a significant elevation in various cathelicidins upon mastitis infection in cows [21,22]. Consistent with prior research, our study identified seven cathelicidins in milk whey, with their abundances correlating with the milk’s SCC. Cathelicidins play a crucial role by exerting direct antibiotic activity, thereby enhancing defense responses and providing protection against pathogens [23]. It is well-established that neutrophil levels rise and tend to increase further with higher SCCs [24,25]. Notably, azurocidin 1 is released when neutrophils arrive at the site of inflammation [26]. Consequently, our study found a significant increase in azurocidin 1 levels with the rise of milk SCC. Azurocidin 1, belonging to the family of serine protease homologs from neutrophils, plays a vital role in immune response processes. This includes not only direct microbicidal activity but also the activation of chemotactic processes that facilitate the recruitment of immune cells to damaged mammary tissue [27,28]. Therefore, we propose that azurocidin 1 could potentially function as a biomarker for diagnosing intramammary infections. β-defensins, known for their role as endogenous antibiotic peptides, have been extensively documented for their expression in the mammary gland [29]. Research has demonstrated that several β-defensin mRNAs, including lingual antimicrobial peptide, β-defensin 4, and β-defensin 5, show elevated levels in mastitic quarters compared to healthy udder quarters [30,31]. However, identifying β-defensin gene products in the mammary gland or milk has historically been challenging. In our study, two β-defensin proteins, β-defensin 2 and β-defensin 12, were observed to increase in milk whey concurrent with rising SCCs. Peptidoglycan recognition protein 1, a conserved pattern recognition molecule, binds to murein peptidoglycans of both Gram-positive and Gram-negative bacteria and is involved in direct microbicidal activity [32]. Additionally, gene polymorphisms in peptidoglycan recognition protein 1 have been linked to mastitis resistance [33]. This protein has also been shown to be upregulated in bovine whey during experimental mastitis infection induced by *Streptococcus uberis* [34]. Collectively, our findings expand the repertoire of novel microbicidal substances that play a role in safeguarding the mammary gland against pathogens. The augmented expression of these antibacterial substances in milk from infected udders directly neutralizes pathogens and amplifies the immune response, facilitating the restoration of normal mammary function.

Our research also revealed that various new actin-related proteins are associated with the milk SCCs. Tropomyosin-1, part of the actin filament binding protein family, has been previously identified in bovine mammary tissue [35]. This protein is implicated in morphological alterations and cell motility and is increasingly recognized as a novel tumor suppressor in the context of breast cancer cellular transformation [36]. Tropomyosin-1 level was observed to decrease with increasing SCC, indicative of morphological damage of mammary cells. Desmin, a crucial intermediate filament protein, plays a vital role in maintaining the structural integrity and mechanical strength of the vascular wall [37]. Therefore, we hypothesize that alterations in these actin-related proteins compromise the structural integrity of mammary cells and the blood–milk barrier. Consequently, this led to the observation that several blood-origin proteins, such as plasminogen, serotransferrin, and kininogen-2, showed increased presence in milk whey. 

Our study demonstrated that the levels of several enzymes were notably altered in accordance with SCCs. This includes enzymes such as serine dehydratase/threonine deaminase, serpin B4, and transketolase, which showed increased levels, and protein disulfide-isomerase A6, which exhibited a decrease. Serine dehydratase/threonine deaminase is known for catalyzing the dehydration of serine and threonine into pyruvate and ammonia. Interestingly, a previous study reported a decrease in the percentage levels of several amino acids, including serine and threonine, in mastitis-affected milk compared to control milk [38]. Furthermore, increased ammonia concentration has been documented in inflamed human breast tissue due to mastitis [39]. Therefore, we infer that the elevation in serine dehydratase/threonine deaminase levels may lead to heightened degradation of serine and threonine, resulting in their diminished abundance and a corresponding increase in the metabolic byproduct, ammonia. Transketolase, a thiamine diphosphate-dependent enzyme, is responsible for catalyzing the transfer of a two-carbon ketol unit from a donor ketose to an acceptor aldose in the non-oxidative branch of the pentose phosphate pathway [40]. Transketolase has been identified in mammary glands and is known to be elevated in mammary gland tumors compared to normal mammary glands [41,42]. We suggest that the increased levels of transketolase in milk may originate from mammary glands infected with pathogens, implying that transketolase could potentially serve as a biomarker for diagnosing intramammary infections.

Additionally, numerous proteins that correlated with the increase in the SCCs are implicated in the inflammatory response. It is also well established that haptoglobin, as an acute-phase protein, increases significantly in response to inflammation and infection in the mammary gland, serving as a reliable indicator of udder health. Notably, α2-macroglobulin, which plays a critical role in multiple biological functions, acts as a physiological protector [43]. A prior study reported a significant increase in α2-macroglobulin levels in milk from cows treated with lipopolysaccharide [44]. As a broad-spectrum protease inhibitor with the ability to inhibit biofilm formation [45], α2-macroglobulin may play a vital role in defending the mammary gland against various pathogens. Kininogen, primarily synthesized in the liver and found in the blood, has also been detected in bovine milk [46]. A previous study revealed that reduced levels of kininogen-1 in the milk of cows on a diet low in neutral detergent fiber and high in rumen fermentable starch were linked to immune suppression [47]. In addition to kininogen-1, which has been previously associated with intramammary infections [48,49], our study also identified a correlation between the expression of kininogen-2 and the SCCs. Kininogen-2, a key component of the plasma kallikrein–kinin complex, produces bradykinin upon cleavage, which in turn initiates inflammation [50]. We suggest that the elevation in kininogen levels in milk is indicative of a compromised blood–milk barrier and could also be considered as a potential biomarker for an inflammatory response. 

## 5. Conclusions

This study characterized the whey protein components associated with SCCs ranging from 7 to 200 × 10⁴ cells/mL using a label-free proteomics approach augmented by the SRM method. The S1 and S2 group exhibited the most similarities, in contrast to the S3–S5 groups. The results of this study indicate that the abundance of specific whey proteins is closely correlated with varying SCC levels, thereby providing critical insights into the intricate relationship between SCC and milk protein composition, which may reflect milk quality. As well, elevated SCC is associated with an increase in the abundance of several proteins, such as azurocidin 1, cathelicidin-4, and peptidoglycan recognition protein 1, which suggest an active immune defense mechanism in the mammary gland. These findings provide a foundation for the identification of potential biomarkers for the early detection of udder health issues, contributing to the development of strategies to produce high-quality milk.

## Figures and Tables

**Figure 1 animals-15-00675-f001:**
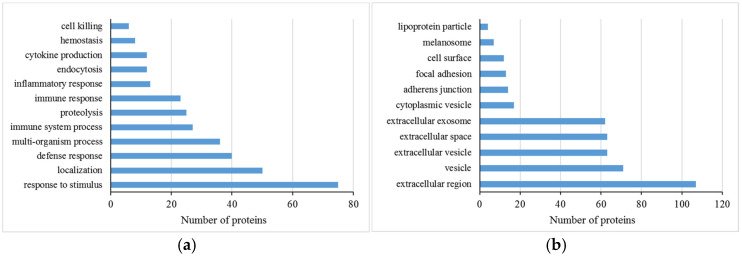
Biological processes (**a**) and cellular components (**b**) of differentially abundant proteins of milk whey among the different somatic cell count milk groups predicted using the DAVID functional annotation tool.

**Figure 2 animals-15-00675-f002:**
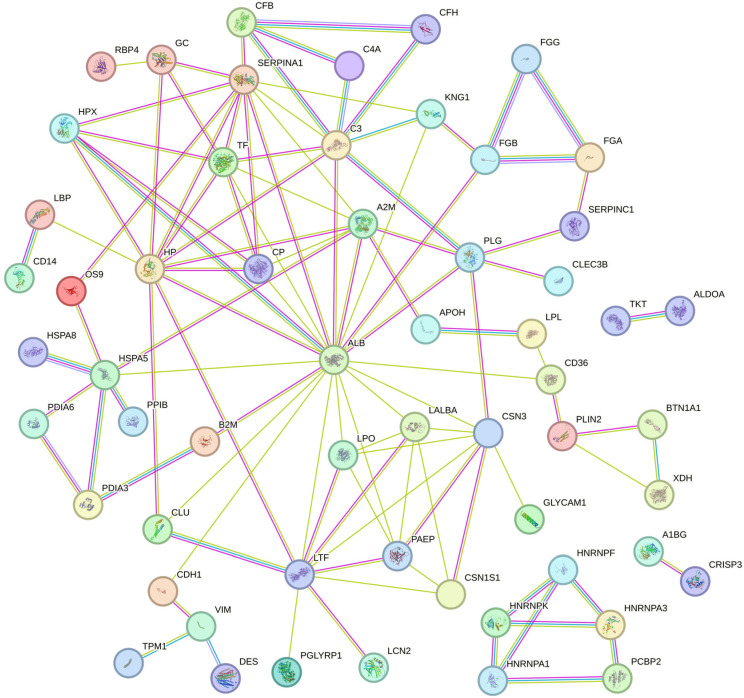
The constructed protein–protein interaction network of differentially abundant proteins in milk whey among the different somatic cell count milk groups. A1BG—Alpha-1-B Glycoprotein; ALDOA—Fructose-Bisphosphate Aldolase; ALB—Albumin; APOH—Apolipoprotein H; A2M—Alpha-2-Macroglobulin; B2M—Beta-2-Microglobulin; BTN1A1—Butyrophilin Subfamily 1 Member A1; C3—Complement Component 3; C4A—Complement Component 4A; CD14—Monocyte Differentiation Antigen CD14; CD36—Platelet Glycoprotein 4; CDH1—Cadherin 1; CFB—Complement Factor B; CFH—Complement Factor H; CLEC3B—Tetranectin; CLU—Clusterin; CP—Ceruloplasmin; CRISP3—Cysteine Rich Secretory Protein 3; CSN1S1—Alpha-S1-Casein; CSN3—Kappa Casein; DES—Desmin; FGA—Fibrinogen Alpha Chain; FGB—Fibrinogen Beta Chain; FGG—Fibrinogen Gamma Chain; GC—Vitamin D Binding Protein; GLYCAM1—Glycosylation Dependent Cell Adhesion Molecule 1; HNRNPA1—Heterogeneous Nuclear Ribonucleoprotein A1; HNRNPF—Heterogeneous Nuclear Ribonucleoprotein F; HNRNPK—Heterogeneous Nuclear Ribonucleoprotein K; HPX—Hemopexin; HSPA5—Endoplasmic Reticulum Chaperone BiP; HSPA8—Heat Shock Cognate 71 kDa Protein; KNG1—Kininogen 1; LALBA—Alpha Lactalbumin; LBP—Lipopolysaccharide Binding Protein; LCN2—Lipocalin 2; LPL—Lipoprotein Lipase; LPO—Lactoperoxidase; OS9—Protein OS-9; PAEP—Beta-Lactoglobulin; PCBP2—Poly(rC) Binding Protein 2; PDIA3—Protein Disulfide-Isomerase A3; PDIA6—Protein Disulfide-Isomerase A6; PGLYRP1—Peptidoglycan Recognition Protein 1; PLG—Plasminogen; PLIN2—Perilipin 2; PPIB—Peptidylprolyl Isomerase B; RBP4—Retinol Binding Protein 4; SERPINA1—Alpha-1-Antiproteinase; SERPINC1—Antithrombin-III; TF—Transferrin; TKT—Transketolase; TPM1—Tropomyosin 1; VIM—Vimentin; XDH—Xanthine Dehydrogenase/Oxidase.

**Figure 3 animals-15-00675-f003:**
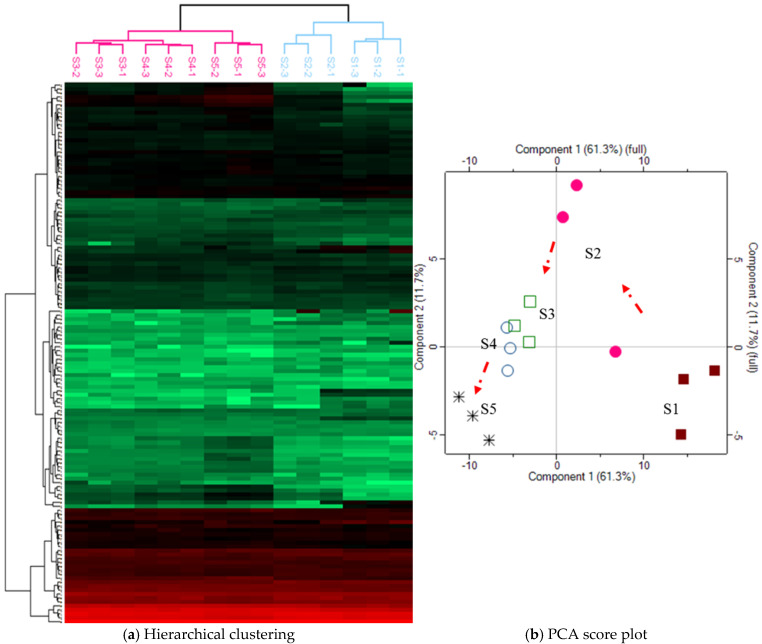
Hierarchical clustering (**a**), and PCA score plot (**b**) of differential proteins among the somatic cell count groups. S1 to S5 groups represent somatic cell counts of 6–9, 17–20, 38–42, 68–80, and 176–243 × 10⁴ cells/mL, respectively.

**Table 1 animals-15-00675-t001:** The results of selected whey proteins quantified by the selected reaction monitoring method.

Accession No.	Protein Name	Selected Peptide Sequence	S1 Group ^1^	S2 Group ^2^	S3 Group ^2^	S4 Group ^2^	S5 Group ^2^	S1/S2 ^3^	S1/S3 ^3^	S1/S4 ^3^	S1/S5 ^3^	−Log *p* Value ^4^
G3MYN2	Interleukin 18 binding protein	LWEGSTR	7.75 × 10^5^	1.37 × 10^6^	1.71 × 10^6^	2.38 × 10^6^	3.23 × 10^6^	0.57	0.45	0.33	0.24	5.41
G3N0Q8	Azurocidin 1	ARPQELPFLASIQNQGR; GTDFFAR	4.79 × 10^5^	2.60 × 10^6^	5.92 × 10^6^	8.51 × 10^6^	2.13 × 10^7^	0.19	0.08	0.06	0.02	6.79
O62654	Desmin	VELQELNDR	8.12 × 10^6^	1.15 × 10^6^	3.87 × 10^5^	4.20 × 10^5^	3.76 × 10^5^	7.07	20.95	19.34	21.57	1.33
P22226	Cathelicidin-1	AVDQLNEQSSEPNIYR	3.86 × 10^6^	1.79 × 10^7^	2.59 × 10^7^	4.58 × 10^7^	8.47 × 10^7^	0.22	0.15	0.08	0.05	8.44
P33046	Cathelicidin-4	AVDQLNELSSEANLYR; TIQQPAEQCDFK	1.73 × 10^6^	6.56 × 10^6^	1.55 × 10^7^	2.55 × 10^7^	6.87 × 10^7^	0.26	0.11	0.07	0.02	6.94
P54228	Cathelicidin-6	TSQQPAEQCDFK	5.54 × 10^5^	2.44 × 10^6^	4.82 × 10^6^	8.01 × 10^6^	1.90 × 10^7^	0.23	0.12	0.07	0.03	8.06
P54229	Cathelicidin-5	TSQQSPEQCDFK; YGPIIVPIIR	3.93 × 10^5^	1.90 × 10^6^	3.16 × 10^6^	6.52 × 10^6^	1.13 × 10^7^	0.21	0.13	0.07	0.04	5.44
Q0VCW4	Serine dehydratase/ threonine deaminase	LVTLPCITSVAK	2.89 × 10^5^	4.36 × 10^5^	3.59 × 10^5^	4.94 × 10^5^	1.73 × 10^6^	0.66	0.80	0.58	0.17	2.55
Q1JPB0	Leukocyte elastase inhibitor	VLELPYEGK; IEQQLTLEK	5.61 × 10^5^	1.52 × 10^6^	2.16 × 10^6^	3.02 × 10^6^	5.20 × 10^6^	0.34	0.23	0.15	0.08	5.48
Q3ZC00	Lymphocyte cytosolic protein 1	AYYHLLEQVAPK	0.00	0.00	0.00	1.52 × 10^5^	4.78 × 10^5^	0.00	0.00	0.00	0.00	2.97
Q8SPP7	Peptidoglycan recognition protein 1	QAQNVQYYHVR; DVQQTLSPGDELYK	6.37 × 10^6^	2.33 × 10^7^	4.31 × 10^7^	6.07 × 10^7^	1.39E+08	0.28	0.16	0.11	0.05	6.82

^1^ The relative abundance (peak area) of target proteins in the first group, which serves as the baseline or reference group in the experimental design. ^2^ The relative abundance (peak area) of target proteins in the second to fifth groups, representing different experimental conditions. ^3^ The ratio of protein abundance in S1 compared to other groups (S2, S3, S4, and S5), used to assess relative changes in protein abundance between groups. ^4^ The statistical significance level, expressed as the negative logarithm of the p-value. A larger -Log p value indicates a more significant difference in protein abundance between groups.

**Table 2 animals-15-00675-t002:** Pathway analysis of differentially abundant whey proteins from the different somatic cell count milk groups.

Pathway Name	Counts ^1^	Percent (%) ^2^	*p* Value ^3^	Fold Enrichment ^4^
Complement and coagulation cascades	13	9.85	1.22E-12	19.51
Staphylococcus aureus infection	7	5.30	1.21E-05	13.17
Salivary secretion	7	5.30	8.59E-05	9.36
Antigen processing and presentation	6	4.55	5.05E-04	8.88
Protein processing in endoplasmic reticulum	7	5.30	0.004	4.60
Spliceosome	6	4.55	0.006	5.16
Biosynthesis of amino acids	4	3.03	0.025	6.26

^1^ The number of differentially abundant proteins associated with each pathway. ^2^ The percentage of differentially abundant proteins in the pathway relative to the total number of significant proteins. ^3^ The statistical significance of pathway enrichment, where smaller values indicate more significant enrichment. ^4^ The ratio of the observed number of proteins in a pathway to the expected number based on a random distribution, representing the degree of pathway enrichment.

## Data Availability

The data used to support the findings of this study are available from the corresponding author upon request.

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
