# Peer review of "Alterations in Whey Protein Abundance Correlated with the Somatic Cell Count Identified via Label-Free and Selected Reaction Monitoring Proteomic Approaches"

_animals, 2025, doi:10.3390/ani15050675_

Round 1
Reviewer 1 Report
Comments and Suggestions for Authors
1- General Information:
This manuscript contains very interesting information about the relationship between somatic cell count and whey proteins in milk. However, it would be better to mention whey protein in the abstract, which correlates highly with the cell count in milk and can be a good parameter for assessing udder health.
2- Some comments on the manuscript:
- Line 38 states: “The somatic cell count in milk is considered the gold standard parameter in the dairy industry for monitoring udder health and milk quality. How can it be explained that some milk samples do not contain pathogenic germs despite very high cell counts?
- In line 73 it was mentioned that milk samples were taken from individual cows. The question is: How is the milk sample taken before using the milking machine, from the first stripping or from the milk after milking with machine. It is known that the cell count of milk samples depends on the time of sampling.
- In the material and methods: To ensure that the milk samples used clear, it is necessary to create a plan for the number and distribution of milk samples.
- On line 93 it is better to write the Bradford kit analysis method and indicate the kits used, since there are two types of kits.
- In line 144, it was mentioned that the data was analysed using ANOVA. However, the authors did not mention how the data was analysed and which statistical program was used.
- In the “Results” section, Table 1 was not mentioned in the text.
- In Table 1 and Note 3, the results of S1/S2, S1/S3, S1/S4 and S1/S5 for the protein desmin were very high compared to other proteins. Therefore, please check the written results in the table.
- The reference list is in order, only the serial number and page number are missing for Nr. 46.
Author Response
Comments 1: [This manuscript contains very interesting information about the relationship between somatic cell count and whey proteins in milk. However, it would be better to mention whey protein in the abstract, which correlates highly with the cell count in milk and can be a good parameter for assessing udder health.]
Response 1: We sincerely appreciate your insightful suggestion. We have added relevant information about whey protein to the abstract. [“The somatic cell count (SCC) is widely used to assess milk quality and diagnose intramammary infections. Several whey proteins have been shown to correlate significantly with SCC and are considered potential indicators of udder health. However, the relationship between milk whey proteins and SCC has not been fully elucidated.” This change can be found in the revised manuscript on page 1, paragraph 1, lines 24-26.]
Comments 2: [Line 38 states: “The somatic cell count in milk is considered the gold standard parameter in the dairy industry for monitoring udder health and milk quality. How can it be explained that some milk samples do not contain pathogenic germs despite very high cell counts?]
Response 2: We have provided an explanation and welcome any feedback or suggestions for improvement. There are several factors that influence somatic cell count (SCC) in milk, which can be categorized as follows: 1)Mammary Gland Health: The primary factor affecting SCC is the health of the mammary gland. Mastitis, particularly clinical mastitis, is the most common cause of elevated SCC. When the mammary gland is infected by bacteria, an immune response leads to the influx of white blood cells, especially neutrophils, into the gland, thereby increasing SCC. 2) Mammary Gland Trauma or Stimulation: External injuries, such as nipple damage or mechanical stress/stretching of the mammary gland, can induce local inflammatory responses, resulting in an increase in SCC. These conditions are typically associated with infections of the mammary gland. 3)Stress Factors: Various stressors, such as heat stress or sudden fear, can also contribute to an increase in SCC. Stress negatively impacts the immune system, which can lead to an elevated response in the mammary gland, thus increasing SCC.
Milk samples exhibit very high somatic cell counts without the presence of pathogenic microorganisms, this phenomenon may be related to the factors mentioned in points 2 and 3 above.
Comments 3: [In line 73 it was mentioned that milk samples were taken from individual cows. The question is: How is the milk sample taken before using the milking machine, from the first stripping or from the milk after milking with machine. It is known that the cell count of milk samples depends on the time of sampling.]
Response 3: The cows were milked using a milking machine equipped with a diverter. We collected a well-mixed milk sample from the diverter, which, in principle, separates the milk from the first to the last drop during the milking process for each cow.
Comments 4: [In the material and methods: To ensure that the milk samples used clear, it is necessary to create a plan for the number and distribution of milk samples.]
Response 4: Thank you for highlighting the need for more detailed information regarding the milk samples. We propose adding the following detailed information about the milk samples:
[In this study, the selected farm, located near Chuzhou City in Anhui Province, China, houses over 600 lactating Holstein cows, with DHI testing conducted on milk samples 1-2 times per month. Prior to sample collection, a preliminary selection of cows was performed based on DHI data, focusing on cows with 2-3 parities, similar milk yields, and in mid-to-late lactation. Additionally, these cows were fed a uniform diet. Subsequently, cows were milked automatically using a milking system equipped with a diverter, with milk yield recorded and milk samples collected from the diverter for further analysis. A total of 100 milk samples (50 mL×2) were purposefully collected from individual dairy cows. These changes can be found in the revised manuscript on page 3, paragraphs 2, lines 95–103.]
Comments 5: [On line 93 it is better to write the Bradford kit analysis method and indicate the kits used, since there are two types of kits.]
Response 5: Thank you for your valuable comments and suggestions on our paper. [Sentences have been revised as: The liquid phase of the milk whey was subsequently collected, and the protein concentration was determined using a Bradford kit (Beyotime Biotechnology, Shanghai, China). In this procedure, the sample was mixed with Coomassie Brilliant Blue G-250 dye, and absorbance was measured at 595 nm. The absorbance value is directly proportional to the protein concentration, with bovine serum albumin used as the standard. These changes can be found in the revised manuscript on page 3, paragraphs 3, lines 121-126.]
Comments 6: [In line 144, it was mentioned that the data was analysed using ANOVA. However, the authors did not mention how the data was analysed and which statistical program was used.]
Response 6: Thank you for your valuable comments and suggestions on our paper. [We have provided a brief description of data processing using Perseus software, which has been added to the document. “The proteins derived from various milk groups were imported, categorized, and preprocessed, including log2 transformation and handling of missing values, followed by statistical analysis via ANOVA, as integrated within Perseus software (www.perseus-framework.org), with significance determined by P values less than 0.05. Additionally, hierarchical clustering and principal component analysis of the quantified proteins were conducted using Perseus software.” These changes can be found in the revised manuscript on page 5, paragraphs 3, Lines 202-207.]
Comments 7: [In the “Results” section, Table 1 was not mentioned in the text.]
Response 7: Thank you for the clarification! Table 1 is cited in line 235.
Comments 8: [In Table 1 and Note 3, the results of S1/S2, S1/S3, S1/S4 and S1/S5 for the protein desmin were very high compared to other proteins. Therefore, please check the written results in the table.]
Response 8: Thank you for the reviewer’s thorough examination. We have carefully verified the data for desmin protein in Table 1. The calculated ratios for S1/S2 (7.07), S1/S3 (20.95), S1/S4 (19.34), and S1/S5 (21.57) are accurate. These higher ratios reflect the high expression of desmin protein in the S1 group, with a significant decline from S1 to S3, and then maintaining a relatively stable level from S3 to S5. This expression pattern has been validated using two complementary proteomics methods, label-free and SRM, and is consistent with the biological function of desmin as a cytoskeletal protein in maintaining the structural integrity of mammary tissue, as described in the discussion section of the paper (Desmin, a crucial intermediate filament protein, plays a vital role in maintaining the structural integrity and mechanical strength of the vascular wall [37], Page 12, Paragraph 1, Lines 356-358). Therefore, we believe the data in Table 1 are accurate and reliable.
Comments 9: [The reference list is in order, only the serial number and page number are missing for Nr. 46.]
Response 9: Thank you for your feedback. [We have added the serial number and page number for reference No. 47. This change can be found in the revised manuscript on lines 542–543.]

Reviewer 2 Report
Comments and Suggestions for Authors
Some potential flaws and limitations must be considered:
- Introduction must be expanded to cover all the aspects of the paper.
- Material a methods also.
- The study analyzed a total of 100 milk samples, divided into five groups based on SCC levels. However, the sample size within each group (12 samples) may be relatively small, which could limit the statistical power and generalizability of the results. Additionally, the samples were collected from a single dairy farm, which may not represent the broader population of dairy cows. Also, this was a one cow – one sample approach, milk from all 4 quarters can differ very significantly.
- The study appears to be cross-sectional, assessing milk samples at a single point in time. Longitudinal studies that track changes in protein profiles over time, especially during different stages of lactation or in response to treatment for mastitis, could provide more comprehensive insights into the dynamics of whey proteins and SCC. It is well known that the composition of milk is not the same through the whole lactation.
- The study does not seem to account for other factors that could influence protein levels and SCC, such as the cow's age, breed, nutrition, and overall health status. These variables could confound the relationship between SCC and whey protein profiles, making it difficult to draw definitive conclusions.
- While the label-free proteomics approach is robust, it may have limitations in sensitivity and quantification accuracy compared to other methods, such as isotope labeling. This could affect the detection of low-abundance proteins that may be relevant to SCC and milk quality.
- The functional analysis of proteins was conducted using DAVID Functional Annotation Tools, which may not capture all biological pathways or interactions. A more comprehensive approach, possibly integrating additional bioinformatics tools, could enhance the understanding of the functional implications of the identified proteins.
Author Response
Comments 1: [Introduction must be expanded to cover all the aspects of the paper.] |
Response 1: Thank you for pointing this out. We agree with this comment. [Therefore, in the introduction, we have expanded the discussion to include relevant literature on label-free proteomics and targeted proteomics. These changes can be found in the revised manuscript on page 1, paragraph 3, lines 41–47; page 2, paragraph 1, lines 53–56; page 2, paragraph 3, lines 70–74; page 2, paragraph 3, lines 79–84; page 3, paragraph 1, lines 87–92.] |
Comments 2: [Material a methods also.] |
Response 2: Thank you for pointing this out. We agree with this comment. Therefore, we have expanded and supplemented the content of the material and methods. [In the Materials and Methods section, we have specified the selection criteria for the lactating cows. The study included Chinese Holstein cows with 2-3 parities, all in mid-to-late lactation, and fed a uniform diet. These changes can be found in the revised manuscript on page 3, paragraphs 2, lines 95–103. Peptides selected for SRM were based on label-free proteomic identification, with a focus on those that were unique to the target protein, exhibited no missed cleavage sites during digestion, were 7-20 amino acids in length, and were free from post-translational modifications. These changes can be found in the revised manuscript on page 5, paragraphs 2, lines 182–185.] |
Comments 3: [The study analyzed a total of 100 milk samples, divided into five groups based on SCC levels. However, the sample size within each group (12 samples) may be relatively small, which could limit the statistical power and generalizability of the results. Additionally, the samples were collected from a single dairy farm, which may not represent the broader population of dairy cows. Also, this was a one cow – one sample approach, milk from all 4 quarters can differ very significantly.] |
Response 3: Thank you for your valuable comments regarding our sample size and sampling methodology. [The experiment was conducted as follows: the selected farm houses over 600 lactating Holstein cows, with DHI testing performed on milk samples 1-2 times per month. Prior to sample collection, we conducted a preliminary selection of cows based on DHI data, focusing on cows with 2-3 parities, similar milk yields, and in mid-to-late lactation (150-250 days). Additionally, these cows were fed a uniform diet. Subsequently, we purposefully collected milk samples from 100 cows. Due to the relatively low number of cows with high somatic cell counts, we expanded the selection criteria to include a broader range of lactation days. This study represents a preliminary analysis of the relationship between somatic cell count and the variation in whey proteins in milk. These changes can be found in the revised manuscript on page 3, paragraphs 2, lines 95–103.] In response to the use of multiple farms, we have considered this issue. In a separate experiment, we collected milk samples from three different farms and are currently conducting metagenomic, metabolomic, and proteomic analyses. Furthermore, several research teams have examined the effects of pathogen infections, such as Staphylococcus aureus, Escherichia coli, and Streptococcus agalactiae, on increases in somatic cell count and mammary gland protein synthesis. However, the observed increases in somatic cell count and changes in milk composition on dairy farms are likely the result of infections caused by a combination of pathogens. Therefore, in this experiment, we categorized milk samples based on somatic cell count to investigate the relationship between whey proteomics and somatic cell count variations, providing a foundation for milk quality control. |
Comments 4: [The study appears to be cross-sectional, assessing milk samples at a single point in time. Longitudinal studies that track changes in protein profiles over time, especially during different stages of lactation or in response to treatment for mastitis, could provide more comprehensive insights into the dynamics of whey proteins and SCC. It is well known that the composition of milk is not the same through the whole lactation.] |
Response 4: We sincerely appreciate your insightful suggestion. In our previous studies, we have examined changes in DHI parameters and protein profiles over time in healthy cows. As the reviewer correctly noted, both DHI parameters and protein profiles fluctuate throughout the lactation period. However, our current research did not focus on the variations at different stages of lactation or in response to mastitis treatment. We recognize the importance of these aspects and intend to incorporate them in future studies, which will provide more comprehensive insights into the dynamics of whey proteins and SCC throughout the different stages of lactation. The present study, being exploratory in nature, offers initial findings on the relationship between whey proteins and SCC and serves as a foundational basis for subsequent investigations. |
Comments 5: [The study does not seem to account for other factors that could influence protein levels and SCC, such as the cow's age, breed, nutrition, and overall health status. These variables could confound the relationship between SCC and whey protein profiles, making it difficult to draw definitive conclusions.] |
Response 5: Thank you for your useful insight on this matter. We have provided a clearer and more detailed description of the experimental conditions. [In this study, the selected farm, located near Chuzhou City in Anhui Province, China, houses over 600 lactating Holstein cows, with DHI testing conducted on milk samples 1-2 times per month. Prior to sample collection, a preliminary selection of cows was performed based on DHI data, focusing on cows with 2-3 parities, similar milk yields, and in mid-to-late lactation. Additionally, these cows were fed a uniform diet. Subsequently, a total of 100 milk samples (50 mL×2) were purposefully collected from individual dairy cows. While the composition of the diet was carefully documented, we did not include the formulation in the study as it was based on the recorded components. Lines 95-103. These changes can be found in the revised manuscript on page 3, paragraphs 2, lines 95–103.] |
Comments 6: [While the label-free proteomics approach is robust, it may have limitations in sensitivity and quantification accuracy compared to other methods, such as isotope labeling. This could affect the detection of low-abundance proteins that may be relevant to SCC and milk quality.] |
Response 6: Thank you for this critical methodological observation. The relationship between SCC and whey protein profiles indeed presents a complex web of biological interactions that require careful experimental control. As your consideration, the label-free proteomics approach and labeled proteomics methods are both widely utilized strategies in mass spectrometry-based proteomics, each offering distinct advantages and limitations. Label-free proteomics is a simpler and more cost-effective approach that does not require expensive reagents, making it suitable for a broad range of samples and high-throughput studies. However, it may face challenges related to sensitivity and quantification accuracy, particularly for low-abundance proteins. In contrast, labeled proteomics techniques, such as TMT or iTRAQ, provide enhanced accuracy and reproducibility in protein quantification, with superior sensitivity for detecting low-abundance proteins. Nevertheless, these methods are more costly, technically complex, and may introduce potential biases due to the labeling process. In this study, we opted for the label-free approach, as it facilitates high-throughput analysis of a larger number of samples, which is particularly advantageous for time-efficient experimental design. To mitigate concerns regarding quantification accuracy inherent in label-free proteomics, we incorporated the SRM (selected reaction monitoring) method to validate the differentially expressed proteins. While label-free quantification may exhibit reduced sensitivity for low-abundance proteins compared to labeled approaches, which could impact the accuracy of detection, the present study serves as an initial exploration into the relationship between whey proteins and SCC. In future investigations, we plan to incorporate labeled proteomics techniques to further examine this relationship. |
Comments 7: [The functional analysis of proteins was conducted using DAVID Functional Annotation Tools, which may not capture all biological pathways or interactions. A more comprehensive approach, possibly integrating additional bioinformatics tools, could enhance the understanding of the functional implications of the identified proteins.] |
Response 7: We sincerely appreciate your insightful suggestions. In response to the feedback, we have included an analysis of protein-protein interactions using the STRING software in this revision. This approach helps to highlight the potential interactions between the differentially expressed proteins, providing a deeper understanding of their functional implications. [The following sentences have been incorporated into the result section. "The PPI network of 129 proteins was extracted from the String database (Figure 2), The interaction sources included text mining, experimental data, and curated databases. A minimum required interaction score of 0.7, indicating high confidence, was applied to select significant protein interactions. Four proteins, including serum albumin, haptoglobin, alpha-1-antiproteinase, and lactoferrin, were identified as hub proteins by screening the nodes with a degree of connectivity greater than 10. Notably, both haptoglobin and lactoferrin are implicated in the defense response. Details of the PPI network are provided in Table S3”. These changes can be found in the revised manuscript on page 9, paragraph 1, lines 272-279.] |

Reviewer 3 Report
Comments and Suggestions for Authors
The work is an interesting approach to research on the possibilities of more precise diagnosis of qualitative changes in milk. It is a valuable contribution to the current knowledge on this subject. However, in my opinion, it requires corrections.
Detailed notes:
L2-4 (The title): The title should be changed. In its current form it is more of a statement of fact. In this form it suggests changes in proteins. However, the manuscript concerns changes in the proportion of protein content in whey.
L24, 31, 35, 60 .... : milk whey proteome: This term refers to the state of proteins found inside cells. The word "proteome" should not be applied to the composition of whey. Must be changed throughout the manuscript.
Introduction: In my opinion, changes in whey protein content should not be directly linked to the number of somatic cells (SCC). This is a great marker of cow udder health. However, changes in SCC and the proportion and quality of milk proteins should be primarily linked to the etiology of inflammation of the mammary gland mastitis. Changes in SCC are a function of the duration of infection and the course of the inflammatory reaction. This should be taken into account, because the number of microorganisms directly affects the body's defense reaction and the appearance of SCC. In my opinion, this aspect should be included in the manuscript.
Materials and Methods
L73 Briefly ?
2.1. Sample collection: More detailed information should be provided regarding the sampling technique for analysis. How representative the sample was maintained, whether samples were taken from all milkings, whether milk collection devices (collection technique) were used.
If SCC was analyzed as a milk classification feature, whether the total number of microorganisms was not tested. In my opinion, this information is missing.
There is no information about the productivity of cows, breed, size (lactations completed), etc.
L85-86: The description is unclear. What trials were combined after thawing?
L149: 2.7. Selected reaction monitoring (SRM) validation. Why is this arrangement used? This subsection (2.7) should be placed before 2.6. Data analysis.
L200 - Table 1: How was the information for the Selected Peptide Sequence obtained? Was it from software libraries? Please clarify the description in MM.
L350: ...varying SCCs... : it should be specified in the description that this applies to the tested SCC classes.
Conclusions
L353-359: (L356) ... inflammatory responses, and cytoskeletal dynamics... too much generalization of the results. This was not found. No associations with bacterial counts were demonstrated. (L356-359) On what basis do the Authors claim this? Changes in the structure of milk-producing cells were not analyzed in this study.
L360: ...novel insights into the correlation between milk quality and the SCCs... . The study did not present the correlation coefficient between the indicated features. Therefore, the use of this term is not precise.
Author Response
Comments 1: [The title should be changed. In its current form it is more of a statement of fact. In this form it suggests changes in proteins. However, the manuscript concerns changes in the proportion of protein content in whey.]
Response 1: Thank you for pointing this out. We agree with this comment. Therefore, we have changed the title. [The manuscript title has been modified to: “Alterations in whey protein abundance correlated with the somatic cell count identified via label–free and selected reaction monitoring proteomic approaches ”. This change can be found in the revised manuscript on page 1, lines 2-4]
Comments 2: [milk whey proteome: This term refers to the state of proteins found inside cells. The word "proteome" should not be applied to the composition of whey. Must be changed throughout the manuscript.]
Response 2: We appreciate your insightful comments.
[According to your suggestion, all instances of “milk whey proteome” or “whey proteome” in the text have been modified to “milk whey proteins”. These changes can be found in the revised manuscript on page 1, paragraph 2, lines 26; page 10, paragraph 1, lines 288.]
Comments 3: [In my opinion, changes in whey protein content should not be directly linked to the number of somatic cells (SCC). This is a great marker of cow udder health. However, changes in SCC and the proportion and quality of milk proteins should be primarily linked to the etiology of inflammation of the mammary gland mastitis. Changes in SCC are a function of the duration of infection and the course of the inflammatory reaction. This should be taken into account, because the number of microorganisms directly affects the body's defense reaction and the appearance of SCC. In my opinion, this aspect should be included in the manuscript.]
Response 3: Thank you for your insightful comment regarding the relationship between somatic cell count (SCC) and whey protein changes. We fully agree that changes in whey protein content should not be directly linked to SCC alone, as these changes are primarily driven by the inflammatory response during mastitis. We appreciate this valuable feedback and propose the following modifications to better address this important mechanistic relationship:
[According to your suggestion, several sentences have been incorporated into the introduction section. “It represents the number of somatic cells, primarily leukocytes, present in a milliliter of milk. SCC typically increases in response to infection or inflammation of the udder, often triggered by pathogenic microorganisms. Elevated SCC levels are commonly associated with a decline in milk quality, as they can lead to alterations in milk composition, such as reduced casein content and an increase in whey proteins. Monitoring SCC is essential for maintaining udder health and ensuring milk quality…. SCC serves as a critical tool for detecting and managing mastitis, a condition that can significantly impact milk yield and farm profitability.” These changes can be found in the revised manuscript on page 2, paragraph 1, lines 41-47; page 2, paragraph 1, lines 53-56.]
Comments 4: [Sample collection: More detailed information should be provided regarding the sampling technique for analysis. How representative the sample was maintained, whether samples were taken from all milkings, whether milk collection devices (collection technique) were used.]
Response 4: Thank you for highlighting the need for more detailed information regarding our sample collection methodology. We acknowledge that comprehensive sampling protocols are crucial for ensuring data reliability and reproducibility in milk quality research. We propose adding the following detailed information to the Sample Collection section:
[In this study, the selected farm, located near Chuzhou City in Anhui Province, China, houses over 600 lactating Holstein cows, with DHI testing conducted on milk samples 1-2 times per month. Prior to sample collection, a preliminary selection of cows was performed based on DHI data, focusing on cows with 2-3 parities, similar milk yields, and in mid-to-late lactation. Additionally, these cows were fed a uniform diet. Subsequently, cows were milked automatically using a milking system equipped with a diverter, with milk yield recorded and milk samples collected from the diverter for further analysis. A total of 100 milk samples (50 mL×2) were purposefully collected from individual dairy cows. These changes can be found in the revised manuscript on page 3, paragraphs 2, lines 95–103.]
Comments 5: [If SCC was analyzed as a milk classification feature, whether the total number of microorganisms was not tested. In my opinion, this information is missing.]
Response 5: The total bacterial count was not measured, which is a limitation in our experimental design. Our initial approach involved collecting three samples per group for 16S rRNA gene sequencing in order to determine the bacterial species present in the milk. However, we found significant variation in the bacterial species within the milk samples. As a result, we plan to conduct further experimental studies in the future, comparing the total bacterial count, bacterial species, and somatic cell count using a larger sample size for more comprehensive research.
Comments 6: [There is no information about the productivity of cows, breed, size (lactations completed), etc.]
Response 6: Thank you for highlighting the need for comprehensive information about the experimental animals. [We have provided the following clearer and more detailed description of the experimental conditions: “In this study, the selected farm, located near Chuzhou City in Anhui Province, China, houses over 600 lactating Holstein cows, with DHI testing conducted on milk samples 1-2 times per month. Prior to sample collection, a preliminary selection of cows was performed based on DHI data, focusing on cows with 2-3 parities, similar milk yields, and in mid-to-late lactation. Additionally, these cows were fed a uniform diet.” These changes can be found in the revised manuscript on page 3, paragraphs 2, lines 95–103.]
Comments 7: [The description is unclear. What trials were combined after thawing?]
Response 7: Thank you for your valuable comments and suggestions on our paper. We highly appreciate your feedback and have carefully considered the issues you raised. We provide the following detailed explanation: “There are 12 milk samples for each group: 6–9 (7.2, S1 group), 17–20 (18.4, S2 group), 38–42 (40.4, S3 group), 68–80 (73.6, S4 group), and 176–243 (208.9, S5 group) × 10⁴ cells/mL. Upon thawing, four samples from each group were pooled in equal volumes to obtain three biological replicates.”
Once again, we appreciate your valuable feedback!
Comments 8: [Selected reaction monitoring (SRM) validation. Why is this arrangement used? This subsection (2.7) should be placed before 2.6. Data analysis.]
Response 8: Thank you for your valuable comments and suggestions on our manuscript. We have taken your feedback very seriously and carefully considered the questions you raised. You suggested placing Section 2.7 before Section 2.6. We also recognize that the original logical order was not appropriate. Your suggestion aligns well with the logical sequence of the experimental process, and this revision makes the manuscript more logical and easier for readers to understand. Therefore, we have made the following revisions to the paper:
[The section order has been adjusted, with Section 2.7 'Selected reaction monitoring (SRM) validation' now placed before Section 2.6 'Data analysis,' These changes can be found in the revised manuscript on page 6, paragraphs 3, lines 236–256; page 7, paragraphs 1, lines 257–262.]
We hope that these revisions will provide a clearer understanding of our experimental design. Once again, we appreciate your valuable feedback!
Comments 9: [L200 - Table 1: How was the information for the Selected Peptide Sequence obtained? Was it from software libraries? Please clarify the description in MM.]
Response 9: Thank you for your valuable comments and suggestions on our manuscript. We have taken your feedback very seriously and carefully considered the questions you raised.
According to your suggestion, the following sentence has been incorporated in the text.
[Peptides selected for SRM were based on label-free proteomic identification, with a focus on those that were unique to the target protein, exhibited no missed cleavage sites during digestion, were 7-20 amino acids in length, and were free from post-translational modifications. These changes can be found in the revised manuscript on page 5, paragraph 2, lines 182–185.]
We hope that these changes will eliminate your doubts about peptide selection and provide a clearer understanding of our experimental design.
Thank you again for your valuable comments!
Comments 10: [L350: ...varying SCCs... : it should be specified in the description that this applies to the tested SCC classes.]
Response 10: Thank you for your valuable comments and suggestions on our manuscript. We have taken your feedback very seriously and carefully considered the questions you raised.
[According to your suggestion, we have revised 'varying SCCs' to 'SCCs ranging from 7 to 200 × 10⁴ cells/mL'. This change can be found in the revised manuscript on page 13, paragraph 2, line 404.]
Thank you again for your valuable comments!
Comments 11: [L353-359: (L356) ... inflammatory responses, and cytoskeletal dynamics... too much generalization of the results. This was not found. No associations with bacterial counts were demonstrated. (L356-359) On what basis do the Authors claim this? Changes in the structure of milk-producing cells were not analyzed in this study.]
[L360: ...novel insights into the correlation between milk quality and the SCCs... . The study did not present the correlation coefficient between the indicated features. Therefore, the use of this term is not precise.]
Response 11: Thank you for the reviewer's insightful comments. We fully agree with these points and will revise the manuscript as follows: [Sentences of lines 353-360 were revised as “The results of this study indicate that the abundance of specific whey proteins is closely correlated with varying SCC levels, thereby providing critical insights into the intricate relationship between SCC and milk protein composition, which may reflect milk quality. As well, elevated SCC is associated with an increase in the abundance of several proteins such as azurocidin 1, cathelicidin-4, and peptidoglycan recognition protein 1, which suggest an active immune defense mechanism in the mammary gland. These findings provide a foundation for the identification of potential biomarkers for the early detection of udder health issues, contributing to the development of strategies to product of high-quality milk.” These changes can be found in the revised manuscript on page 13, paragraph 2, lines 406–414.]
Thank you again for your valuable comments!

Round 2
Reviewer 2 Report
Comments and Suggestions for Authors
Thank you extensive corrections.